# New Approaches in Early-Stage NSCL Management: Potential Use of PARP Inhibitors and Immunotherapy Combination

**DOI:** 10.3390/ijms24044044

**Published:** 2023-02-17

**Authors:** Marta Pina Fernandes, Cristina Oliveira, Hugo Sousa, Júlio Oliveira

**Affiliations:** 1Medical Oncology Service, Portuguese Oncology Institute of Porto (IPO Porto)/Porto Comprehensive Cancer Center (Porto.CCC), Rua Dr. António Bernardino de Almeida, 4200-072 Porto, Portugal; 2Early Phase Clinical Trial Unit—IPO Porto/RISE@CI-IPOP (Health Research Network), Portuguese Oncology Institute of Porto (IPO Porto)/Porto Comprehensive Cancer Center (Porto.CCC), Rua Dr. António Bernardino de Almeida, 4200-072 Porto, Portugal; 3Molecular Oncology and Viral Pathology Group, Research Center (CI-IPOP)/RISE@CI-IPOP (Health Research Network), Portuguese Oncology Institute of Porto (IPO Porto)/Porto Comprehensive Cancer Center (Porto.CCC), 4200-072 Porto, Portugal; 4Clinical Pathology Service, Department of Pathology and Laboratory Medicine, Portuguese Oncology Institute of Porto (IPO Porto)/Porto Comprehensive Cancer Center Raquel Seruca (Porto.CCC), Rua Dr. António Bernardino de Almeida, 4200-072 Porto, Portugal; 5Instituto Superior de Saúde—ISAVE, Rua Castelo de Almourol n° 13, 4720-155 Amares, Portugal; 6Department of Biological Sciences, FFUP—Faculty of Pharmacy, University of Porto, Rua de Jorge Viterbo Ferreira n° 228, 4050-313 Porto, Portugal; 7Clinical Pathology Service, Centro Hospitalar de Entre Douro e Vouga, R. Dr. Cândido Pinho 5, 4520-211 Santa Maria da Feira, Portugal; 8Experimental Pathology and Therapeutics Group, Research Center (CI-IPOP)/RISE@CI-IPOP (Health Research Network), Portuguese Oncology Institute of Porto (IPO Porto)/Porto Comprehensive Cancer Center (Porto.CCC), Rua Dr. António Bernardino de Almeida, 4200-072 Porto, Portugal

**Keywords:** non-small cell lung cancer, EARLY-stage NSCLC, PARP inhibitors, immunotherapy, combination therapy, neoadjuvant setting

## Abstract

Lung cancer is the second most common cancer in the world, being the first cause of cancer-related mortality. Surgery remains the only potentially curative treatment for Non-Small Cell Lung Cancer (NSCLC), but the recurrence risk remains high (30–55%) and Overall Survival (OS) is still lower than desirable (63% at 5 years), even with adjuvant treatment. Neoadjuvant treatment can be helpful and new therapies and pharmacologic associations are being studied. Immune Checkpoint Inhibitors (ICI) and PARP inhibitors (PARPi) are two pharmacological classes already in use to treat several cancers. Some pre-clinical studies have shown that its association can be synergic and this is being studied in different settings. Here, we review the PARPi and ICI strategies in cancer management and the information will be used to develop a clinical trial to evaluate the potential of PARPi association with ICI in early-stage neoadjuvant setting NSCLC.

## 1. Non-Small Cell Lung Cancer Management

Lung cancer is the second most common cancer in the world, with an estimated incidence of 2.2 million new cases a year, being also the first cause of cancer-related mortality with approximately 1.8 million deaths a year [1]. Smoking remains the main risk factor, however, environmental exposure to certain substances, such as asbestos, air pollution, chemicals, or genetic alterations can also increase the risk of lung cancer development. Histologically, Lung Cancer, is divided into two distinct groups: Small Cell Lung Cancer (SCLC) and Non-Small Cell Lung Cancer (NSCLC). The latter is the most frequent subtype as it accounts for approximately 85% of all lung cancer cases worldwide [2,3,4] Although the paradigm of treatment for these patients has been suffering several recent alterations, the prognosis remains dismaying as 85% of the patients are diagnosed in advanced stages and treatment options are still limited [1,2,3,5].

The treatment of NSCLC requires the consideration of different variables such as the stage of the disease, histological classification, Programmed Death Ligand-1 (PDL-1) expression, mutational status, and patient performance status. Currently, the NSCLC treatment is based on surgery, radiotherapy, chemotherapy, immunotherapy, and Tyrosine Kinase Inhibitors (TKI) for targeted molecular alterations, either in monotherapy or in combination. In the early stages (I-II), the elective treatment remains surgery either alone or followed by adjuvant treatment with chemotherapy alone or combined with radiotherapy. In locally advanced stage III disease (T3-4, N2-N3), which occurs in approximately 30% of patients with NSCLC, most patients are non-surgical candidates and, currently, the standard of care treatment is chemoradiotherapy followed by maintenance immunotherapy [3,5,6]. In patients with advanced NSCLC, immunotherapy and TKI have been offering a major benefit in clinical outcomes [7,8]. Furthermore, the mutational status has great importance for NSCLC treatment planning, as approximately 10 to 30% harbor an activating mutation in the tyrosine kinase domain of the EGFR gene, most commonly exon 19 deletions or exon 21 L858R mutation; 5% ALK gene rearrangements; 2 to 4% ROS1 or RET rearrangements; 1% to 3% BRAF V600E mutations and less than 1% have NTRK gene fusions [6,7].

Notwithstanding the multiple recent advances in the treatment of advanced NSCLC, early stages remained with the same standard of care for decades [9]. Surgery is considered the only potentially curative treatment for medically fit patients with early stages of NSCLC (stage I, II, and IIIA) offering a 5-year Overall Survival (OS) of nearly 63%. The recurrence rate is dependent on the disease stage at diagnosis. However, adjuvant treatment with cisplatin-based chemotherapy offers only, approximately, 5% benefit in 5-year OS, resulting in a high risk of recurrence (30–55%) and unsatisfactory survival rates at 5 years [6,10,11,12,13]. Due to the encouraging results of biomarker-driven approaches in advanced NSCLC, the neoadjuvant strategies in the treatment of early-stage NSCLC are being revisited. Neoadjuvant treatment allows tumor size reduction(downstaging) with an enhancement of the probability of R0 resection, and micrometastasis control with a noteworthy improvement of disease-free survival (DFS) and OS [9,14,15].

In recent years, several therapies, such as targeted therapy, immunotherapy, poly-ADP-ribose polymerase (PARP) inhibitors, and others, either alone or in combination, are being studied in the neoadjuvant setting some of which with favorable results already available [6,9,14,15,16,17,18].

## 2. Immunotherapy

The theory of immune surveillance suggests that the immune system is able of destroying aberrant cells and preventing cancerous development. Some studies show that patients with compromised immune systems can have an increased risk of developing cancers but also that the cancerous cells can gain the ability to evade the immune response [14,15,18,19,20,21,22,23,24,25,26].

It is known that Programmed Cell Death Protein 1 (PD-1) is highly expressed in Tumor-Infiltrating Lymphocytes (TILs) and that PD-1 Ligands (especially PD-L1) and Cytotoxic T Lymphocyte Associated protein 4 (CTLA-4) can be expressed in many cancerous cell types. PD-L1 can be expressed or upregulated in cancerous cells by two general mechanisms, namely innate resistance and adaptative immune resistance. In innate immune resistance, PD-L1 expression is upregulated by constitutive oncogenic signaling through aberrant activation of the PI3K-AKT pathway or chromosomal alterations and amplifications. PD-L1 expression in tumor cells can also be induced and/or upregulated as a response to inflammatory signals by multiple cytokines, Interferon (INF)-γ being the most potent [8,27,28].

PD-L1/PD-1 interaction in the tumor microenvironment (TME) also promotes T cell dysfunction, exhaustion, apoptosis, neutralization, and production of Interleukin (IL)-10 creating a state of resistance from Cytotoxic T cell (CD8+) mediated tumor cell destruction. All these mechanisms are crucial to hijack the immune system and play a vital role in tumor development, progression metastasis, and survival by escaping tumor immune surveillance [27,28]. Hence, immunotherapy has gained a new major role in cancer therapy by its ability to modulate the adaptative immune response to enhance antitumoral immunity, produce durable responses, and prolong survival in cancer patients [19,22,27,29]. Immunotherapy with Immune Checkpoint Inhibitors (ICI), such as anti-PD-1/PD-L1 and anti-CTLA-4 antibodies, is at the top of interest in cancer research and is now the standard of care for the treatment of several cancers.

The first ICI to be tested and approved for the treatment of cancer patients was the anti-CTLA-4 blocking antibody—Ipilimumab [30,31,32]. CTLA-4 is an immune checkpoint molecule of the B7/CD28 family expressed by different subsets of T cells (e.g., regulatory T cells, activated CD4+ T cells, exhausted T cells) in addition to tumor cells [28,33,34,35].

T-cell activation requires engagement between the T-cell receptor and the Major Histocompatibility Complex (MHC), but also between co-stimulatory receptors on the surface of the T cell with co-stimulatory ligands expressed by Antigen Presenting Cells (APC). CD28 and CTLA-4 are two important T-cell co-stimulatory receptors that have antagonist mechanisms of action [8,31]. CD28 is expressed constitutively on the surface of T cells, after engagement with CD80 (B7-1) or CD86 (B7-2), it augments the T-cell Receptor peptide (TCR)–MHC signal to promote T-cell activation, proliferation, and IL-2 production. On the other hand, CTLA-4 affects IL-2 production, and IL-2 receptor expression and interrupts the cell cycle progression of activated T cells, which leads to the antagonization of T-cell activation. The overall T-cell response is determined by the incorporation of stimulatory and inhibitory signals [28,33,34,35].

In summary, the major function of the anti-CTLA-4 is to allow T-cell activation, proliferation, and migration to the tumor tissue mediating the death of tumor cells [8,28,35,36]. The anti-CTLA-4 mechanism of action is described in Figure 1.

Ipilimumab was first approved in 2011 for the treatment of unresectable or metastatic melanoma as it showed an improvement in OS with manageable toxicities. Currently, it is approved in monotherapy or combination with nivolumab, an anti-PD-1 antibody, in multiple settings (e.g., melanoma, NSCLC, renal cell carcinoma, colorectal carcinoma with Mismatch Repair Deficiency (dMMR) or Microsatellite Instability-High (MSI-H), and others) [28,39]. New anti-CLTA-4 molecules (e.g., tremelimumab) are being studied in several oncologic settings (e.g., NCSCL, SCLC, hepatocarcinoma, etc.) [40,41,42]. The other key immune checkpoint mediator, PD-L1/PD-1, is responsible for the suppression of T cell migration, proliferation, and secretion of cytotoxic mediators, hence restricting tumor cell attack [27,28,43].

PD-1, also known as CD279, is homologous to the CD28 family of protein receptors. PD-1 is predominantly expressed on memory T cells in peripheral tissues after activation by TCR/antigen-loaded MHC and CD28/B7 interactions. It is also vastly expressed on regulatory T cells (TReg) leading to inhibition of immune responses by expression of the forkhead transcription factor FOXP3, absence of expression of effector cytokines (such as IFN-γ), and production of inhibitory cytokines (such as Transforming Growth Factor [TGF]-β, IL-10, and IL-35). Less commonly, PD-1 can also be expressed in B cells, activated monocytes, dendritic cells, and Natural Killer (NK) cells [27,28,43].

PD-L1, also known as CD274 or B7-H1, and PD-L2, also known as CD273 or B7-DC, are two ligands for PD-1 and both members of the B7 family [27,28]. PD-L1 is expressed in many cell types, including APCs, T cells, B cells, monocytes, and epithelial cells. It can be upregulated in many cell types as a response to proinflammatory cytokines (such as Interferon [INF]-γ, IL-4, and activation of transcription-1 (STAT1) and IFN regulatory factor-1 [IRF1]). Contrasting to PD-L1, PD-L2 expression is largely restricted to APCs but it can also be induced on Dendritic Cells (DC), macrophages, and bone marrow-derived mast cells [27,28].

Activation of the PD-1/PD-L1 signaling axis negatively regulates T cell-mediated immune responses by largely reducing cytokine production, namely IFN-γ, Tumor Necrosis Factor (TNF)-α, and IL-2 production. This cascade also has effects on cell differentiation and survival, directly by inhibiting early activation events (positively regulated by CD28) and indirectly through IL-2 [27,28]. PD-1 ligation inhibits the induction of the cell survival factor B-Cell Lymphoma-Extra Large (Bcl-xL) as well as the expression of transcription factors associated with effector cell function, including GATA Binding Protein-3 (GATA-3), T-box transcription factor TBX21 (T-bet) and Eomesodermin/Tbr2 gene (EOMES). In addition, the PD-1 axis also inhibits the apoptotic activity on activated cells, including on B cells and NK cells [27,28].

The inhibition of PD-1/PD-L1 bound by anti-PD-L1 or anti-PD-1 antibodies reverses T-cell suppression and enhances endogenous anti-tumor immunity [8,19,44,45]. Anti-PD-1/PD-L1 mechanism of action is described in Figure 2.

The first anti-PD-1 approved was nivolumab in 2014. During the following 4 years, several other inhibitors of the PD-1 receptor or its ligands, PD-L1 and PD-L2, were approved, with pembrolizumab, atezolizumab, durvalumab, and avelumab showing significant enhancement in OS and Progression Free Survival (PFS) in several cancer types.

Currently, the inhibitors of PD-1/PD-L1 either in monotherapy or combined with chemotherapy are approved in stage IV NSCLC as the first-line setting providing long-lasting responses and survival [46,47]. Furthermore, PD-1/PD-L1 inhibition is already being tested in resectable early-stage NSCLC in a neoadjuvant setting with promising results [14,15,18,48,49,50].

### Neoadjuvant ICI in NSCLC

Lately, several clinical trials, to evaluate the safety and feasibility of neoadjuvant ICIs in NSCLC have been developed. Their results propose that ICIs could be better tolerated than standard neoadjuvant chemotherapy and more effective in reducing cancer recurrence and metastasis [4,6,8,51]. Most of the clinical trials investigating neoadjuvant ICI are still ongoing, and only partial results are available. Current studies are focusing, not only on neoadjuvant immunotherapy in monotherapy but also in combination with other therapies (e.g., with radiotherapy) as a part of a multimodal approach, followed by surgery and, sometimes, adjuvant immunotherapy. The endpoints are heterogeneous and rely on the general efficacy of neoadjuvant immunotherapy, throughout survival surrogates (MPR, Major Pathological response, defined as ≤10% viable tumor cells; pCR, pathological Complete Response), safety, and feasibility of therapy [8,35,51].

CheckMate 816 is a phase III trial, which enrolled 773 patients with stage IB to IIIA resectable NSCLC to receive nivolumab plus platinum-based chemotherapy or platinum-based chemotherapy alone, followed by resection. The median Event-Free Survival (EFS) was 31.6 months with nivolumab plus chemotherapy and 20.8 months with chemotherapy alone. The percentage of patients with a pCR was 24.0% and 2.2%, respectively and irrespective of PD-L1 expression. Grade 3 or 4 treatment-related adverse events (AEs) occurred in 33.5% of the patients in the nivolumab-plus-chemotherapy group and 36.9% of those in the chemotherapy-alone group. Moreover, 83% of patients who received nivolumab underwent surgery and achieved a complete surgical resection (R0). Surgery-related and treatment-related AEs were similar in both arms. In conclusion, this study demonstrated that neoadjuvant chemo-immunotherapy does not affect the feasibility of surgery and increases pCR [16].

Several phase II trials are ongoing to evaluate neoadjuvant ICI in NSCLC:

The **NADIM trial** is an open-label, multicenter, single-arm with patients with surgically resectable stage IIIA NSCLC. These patients received neoadjuvant treatment with paclitaxel plus carboplatin plus nivolumab for three cycles before surgical resection, followed by adjuvant nivolumab monotherapy for 1 year. At 24 months, progression-free survival (PFS) was 77,1%, and 30% had toxic events ≥ grade 3 [52]. 

The **PRINCEPS trial** enrolled 30 patients with resectable clinical stage IA-IIIA NSCLC to be treated with one cycle of induction atezolizumab. None of the patients had delayed surgery and all had complete resection (R0). MPR was not observed, however, this could be explained by the short delay between the infusion of atezolizumab and surgery (between 21 and 28 days). Once again, this trial proved the safety and feasibility of neoadjuvant immunotherapy [53].

The **NCT02716038**, included 30 patients with stage IB-IIIA NSCLC treated with four cycles of atezolizumab plus carboplatin plus nab-paclitaxel in a neoadjuvant setting. The primary endpoint was MPR and it was achieved in 57% of patients [54].

The **NEOMUN trial** included 30 patients with NSCLC stage II/IIIA suitable for curative intent surgery. The patients were treated with two cycles of pembrolizumab, followed by tumor resection. MPR was observed in 27% of the patients with 33% of grade 2-3 related adverse events. In conclusion, neoadjuvant pembrolizumab resulted in a feasible and safe treatment [15].

In **NA_00092076**, 21 patients with untreated, surgically resectable early (stage I-IIIA) NSCLC were treated with two preoperative doses of nivolumab. An MPR was observed in 45% of patients, irrespectively of PD-L1 expression [48].

**TOP 1501** enrolled 35 patients with untreated clinical stage IB to IIIA NSCLC, treated with two cycles of pembrolizumab (followed by surgery. Of all patients who underwent surgery, MPR was observed in 28%. Immunotherapy use was not associated with excess surgical morbidity or mortality [55].

The **NEOSTAR trial** included patients treated with nivolumab or nivolumab plus ipilimumab followed by surgery. A total of 44 patients with operable NSCLC were included. The results showed an MPR of 22% and 24% and a pCR of 10% and 38%, in nivolumab and nivolumab plus ipilimumab, respectively [50].

**SAKK 16/00 trial** enrolled 68 patients with resectable stage IIIA(N2) NSCLC. These patients were treated with three cycles of cisplatin plus docetaxel followed by two doses of durvalumab, surgery, and durvalumab maintenance for 1 year. A total of 55 patients were submitted to surgery, with 62% MPR, 18% pCR, and an EFS of 73% at 1 year [56].

The **NeoTAP01**, a multi-center clinical trial, included 33 patients with stage IIIA or T3-4N2 IIIB NSCLC considered surgically resectable. Patients received three cycles of neoadjuvant treatment with toripalimab plus carboplatin, and pemetrexed (for adenocarcinoma) or nab-paclitaxel (for other subtypes). Surgical resection was performed 4-5 weeks later. MPC was achieved in 60.6% of patients with 45.5% with pCR. Toxicities were manageable [57]. 

**LCMC3 trial** included a single-arm study to investigate the efficacy and safety of atezolizumab monotherapy, as neoadjuvant therapy, in patients with resectable (IB- selected IIIB) NSCLC patients. A total of 181 patients were enrolled, but only 143 were included in the primary efficacy analysis, 20% presented MPR and 6%pCR [58]. 

Some other promising ongoing trials with multimodal or new therapeutic combinations approach are being developed (e.g., SQUAT, AEGEAN, and NeoCOAST trials) [59,60,61]. These data allow us to observe that immunotherapy is effective in the neoadjuvant setting without compromising its safety or surgical timing [62].

## 3. PARP Inhibitors

Normal cells can protect themselves against the harmful effects of Deoxyribonucleic Acid (DNA) damage and keep the integrity of the genome. This genetic stability is maintained through several cellular mechanisms such as regulation of DNA damage signaling, recruiting effector proteins for DNA repair, chromatin remodeling, transcription, and stabilization of replication forks. The actual impact of DNA-damage repair (DDR) gene alterations remain unsatisfactorily explored in oncology, and research concerning its role in the genesis and progression of cancer, and potential therapeutic targets is ongoing, including in lung cancer [63,64,65,66].

Smoking is a major risk factor in lung cancer, it induces DNA damage, and promotes the activation of several repair mechanisms that might offer a rationale for targeting DDR defects in selected lung cancer patients. Even though smoking-related lung cancer is related to a higher tumor mutational burden (TMB), it remains uncertain if specific DDR gene alterations are more common in lung cancer that arises in smokers [66,67]. 

Double-strand DNA breaks (DSB) constitute the most severe kind of DNA damage, as they disrupt both DNA chains, leading to mutations or chromosomal reorganizations, increasing the oncogenic risk, and ultimately causing cell death. Homologous recombination Repair (HRR) represents a crucial mechanism for DSB reparation [68]. HRR is considered a “conservative” mechanism of DNA repair, as it restores the DNA sequence at the site of DNA damage by using a homologous DNA sequence as a guide to repairing DSB. When cells become HRR deficient “non-conservative” forms of DNA repair predominate, such as Non-Homologous End Joining (NHEJ) [65,68]. These, “non-conservative”, are effective in DNA repair, however more error-prone, potentially causing modifications in the DNA sequence and genetic mutations. Breast Cancer gene (BRCA)1 and BRCA2 are critical proteins involved in HRR, though several other genes confer a similar phenotype named “BRCAness”. These include mutations in Ataxia Telangiectasia Mutation (ATM), Ataxia Telangiectasia and Rad3-related (ATR), BRCA1 Associated Ring Domain 1 (BARD1), BRCA1 interacting protein (BRIP1), Checkpoint Kinase 1 (CHK1), Checkpoint Kinase 2 (CHK2), Partner and Localizer of BRCA2 (PALB2), RAD51 Recombinase (RAD51), Fanconi Anemia Complementation group (FANC) and pathogenic variants of HRR genes, as these genetic alterations might confer a phenotype similar to BRCAness [38,68,69,70,71].

These findings explain, at least partially, the reason why alterations in DDR genes are identified in 1/3 of malignancies and raise several questions regarding the role of such deficiencies in lung cancer [65,66,68]. Some series report that the overall frequency of HRR mutations in NSCLC is about 14.2%, the most common mutated genes being ATM, BRCA2, AT-rich interactive domain-containing protein (ARID) 1, CHEK2, BRCA1, ATR, RAD50, MSI, and also high TMB. This suggests that, like other solid tumors, NSCLC might have a subset of patients with DDR deficiencies, which can be more prone to drugs that target DDR alterations. Patients with NSCLC that are DDR mutants are associated with better clinical outcomes when treated with PD-(L)1 blockade, as DDR gene alterations correlate with an increase in tumor-infiltrating lymphocytes, genomic instability, TMB, and PD-L1 expression [67,72]. The existence of defective DDR pathways leads cancer cells to rely on other compensatory DDR mechanisms that could also be further explored as a therapeutic target [66].

Single-strand DNA breaks (SSB) represent another type of DNA damage. SSBs are fixed by three mechanisms: base excision repair (BER), nucleotide excision repair (NER), and mismatch repair (MMR). The vital enzymes to effectively accomplish these processes are from the Poly Adenosine Diphosphate (ADP) Ribose Polymerase (PARP) family. This family includes a group of 17 proteins, of which PARP1-3 are delegated to repair DNA breaks through BER that supply BRCA inefficiency. PARP-1 is the most abundant member of ADP-ribosyl transferases of the PARP family and is responsible for approximately 80–90% of the PARylation activity in cells [63,64,65,68,73]. PARP-1 and PARP-2 regulate the poly-ADP ribosylation of chromatin and auto-PARylation. PARP-1 binds to damaged DNA at SSBs and other DNA-damaged locations, leading to numerous changes in the structure of PARP-1 and activation of its catalytic function opening up chromatin. This prompts PARylation, DNA reparation effectors enrollment (such as BRCA1 for HRR or NHEJ-associated factors), and chromatin structure remodeling around damaged DNA. Lastly, it PARylates itself (auto-PARylation) resulting in the release of PARP from the DNA and allowing DNA-repairing proteins to access DNA and terminate the repair process [65,68]. This fact is of extreme importance, as several drugs that target DDR alterations are already approved in some specific settings and new others are under development [66].

Hence, BRCA mutant cells are HRR inefficient, so if PARP is blocked they are more prone to accumulate SSBs and produce potential DSBs, inducing genomic instability and culminating in cell death, the so-called “synthetic lethality” [65,68]. This characterizes the rationale for PARP inhibitors (PARPi) development, being the first clinically approved drug developed to specifically target the DNA repair mechanisms, especially in cancers with defects in DNA damage repair systems, particularly, *BRCA1* and *BRCA2* mutations [65,74]. Moreover, studies revealed that PARP-1 is meaningfully upregulated in many cancer cell lines and malignant tissues [38,69,73]. Indeed, PARP-1 has gathered significant attention as a therapeutic target since its inhibition can be an effective treatment for individuals who have variants in genes that are involved in DNA repair [38,69,73,74].

One of the most important mechanisms of action of PARPi occurs by entrapment of the PARP-1 protein at the replication fork, blocking PARylation reactions (such as transcription and/or translation) and auto-PARylation, increasing PARP1 eagerness for DNA after allosteric changes in its structure. Resulting in a break in the progression of the replication fork, and conducing to a cytotoxic effect, as unrepaired SSBs convert into DSBs, leading to cellular death [38,68,69]. PARPi mechanism of action is described in Figure 3. Another possible contribution is that PARPi enhances NHEJ, leading to additional genomic instability and cell death, the combo of these two effects is stronger than PARP depletion on its own [68,73].

Olaparib was the first approved PARPi, in 2014, as a maintenance treatment for platinum-sensitive, recurrent serous ovarian cancer patients with a BRCA1/2 mutation who had received at least two previous lines of chemotherapy. This treatment resulted in a noteworthy augmentation in PFS despite not having a significant impact on OS [76,77]. It was shortly followed, in 2017, by the approval of rucaparib in the same set of patients [78]. Since then, other PARPi (e.g., niraparib, talazoparib, and veliparib) were developed and new indications for its use were also approved, such as in breast, prostate, and pancreatic cancer with BRCA 1/2 mutation. In some cancer types, such as ovarian, some studies also showed benefits and led to the approval of PARPi in BRCA 1/2 wild-type or carriers of other pathogenic variants in HRR genes such as loss of heterozygosity or homologous repair deficiency [68,79,80,81,82].

### 3.1. PARPi in NSCLC

Few trials including PARPi in monotherapy were conducted in NSCLC. Some of the most relevant included platinum-sensitive NSCLC patients, however, they did not show improved outcomes [83,84].

Some of the results already available of the use of PARPi in NCLSC are phase II trials. The **PIN trial**, a randomized-control trial, included 70 platinum-sensitive NSCLC patients, irrespective of BRCA mutational status, to receive either olaparib or a placebo. Patients that received olaparib had a longer PFS (16.6 months *versus* 12 months HR 0.83, 80% CI 0.6–1.15, *p* = 0.23), however, it was not statistically significant [83].

In **S1900A, Lung-MAP sub-study**, NSCLC patients had a high genomic loss of heterozygosity or BRCA1/2 mutation, progression on platinum-based or anti-PD-(L)1 therapy, and were treated with rucaparib in monotherapy. This study was precociously closed after an interim analysis demonstrated futility [84]. 

Hence, the presence of mutations and changes in the expression of crucial DDR genes in lung cancer, notwithstanding having a strong rationale for the use of PARPi, it is recognized that this therapy in monotherapy might not be sufficient to correctly address this disease. However, associations with other therapies, such as immunotherapy, might grant better outcomes [66,72,84].

### 3.2. PARPi in the Neoadjuvant Setting

Several trials, in different phases, showed benefits in efficacy and safety of PARPi use in monotherapy or association with chemotherapy in neoadjuvant settings in breast cancer, some of which are: [85,86,87,88,89,90].

Phase I trials: **NCT03329937**, which evaluated neoadjuvant niraparib antitumor activity and safety in 21 patients with localized HER2-negative, BRCA-mutated breast cancer. After two cycles, 90.5% had tumor response (≥30% reduction from baseline) by magnetic resonance imaging (MRI). After 2–6 cycles, 40.0% of the patients had pCR; no new safety signals were identified [86]. The **NCT03499353 trial** enrolled 20 patients with breast cancer stage I to III, HER2 negative, BRCA mutated. They were treated with talazoparib for 6 months followed by definitive surgery. Here, 63% had MPR with 53% of the patients having pCR, with manageable toxicity [88].

Phase II trials: the **PETREMAC trial**, included 222 patients with stage II/III breast cancer, that were stratified to eight different neoadjuvant treatment regimens based on estrogen and progesterone receptors, HER2 expression as well as TP53 mutational status. The 32 patients with TNBC, irrespectively of BRCA and TP53 mutational status, received olaparib monotherapy for up to 10 weeks, followed by neoadjuvant chemotherapy. Based on combined clinical and MRI evaluation, olaparib treatment yielded 1 clinical complete response and 17 partial responses (ORR of 56.3%). Response to olaparib occurred independent of tumor size. Even in the 27 patients not harboring BRCA 1/2 or PALB mutations, the ORR was 51.9%. Olaparib was well tolerated, with only one patient experiencing grade >2 toxicity requiring a dose reduction [85]. The **GeparOla trial** assessed the efficacy of paclitaxel plus olaparib in comparison to paclitaxel plus carboplatin followed by epirubicin and cyclophosphamide, in neoadjuvant treatment in HER-negative patients with early breast cancer with HRD. A total of 107 patients were randomized, with 69 patients to the arm of Olaparib plus paclitaxel. The pCR rate in the arm with olaparib was 55.1% *versus* 48.6% in the chemotherapy-only arm. Significantly fewer patients in the olaparib arm experienced grade 3-4 hematologic toxicities 46.4% versus 78.4% (*p* = 0.002) [87]. In **the I-SPY2 trial**, a combination of durvalumab and olaparib added to standard paclitaxel neoadjuvant chemotherapy (durvalumab/olaparib/paclitaxel—DOP) was investigated in stage II-III HER2-negative breast cancer. Seventy-three participants were randomized to the intervention arm and 299 to standard-of-care (paclitaxel) control. DOP increased pCR in all breast cancer subtypes (in all HER2-negative 20–37%, hormone receptor-positive/HER2-negative 14–28%, and TNBC 27–47%). In the DOP arm, 12.3% of patients had immune-related grade 3 AEs (irAEs) *versus* 1.3% in control [89]. 

In the phase III **BrighTNess trial**, candidates for potentially curative surgery patients with formerly untreated stage II-III TNBC were enrolled. They were randomly assigned (2:1:1) to receive one of three regimens: paclitaxel plus carboplatin plus veliparib; paclitaxel plus carboplatin; or paclitaxel; followed by four cycles of doxorubicin and cyclophosphamide in all patients. A total of 634 patients were randomly assigned: 316 to paclitaxel plus carboplatin plus veliparib, 160 to paclitaxel plus carboplatin, and 158 to paclitaxel alone. pCR was reached in 58% of patients in the paclitaxel plus carboplatin group, 53% of patients in paclitaxel, carboplatin plus veliparib, and 31% in paclitaxel alone. Serious AEs, (≥grade 3) were more common in patients receiving carboplatin, however, veliparib did not significantly augment toxicity. The investigators concluded, that in this study, the addition of carboplatin to paclitaxel improved pCR, however, the addition of veliparib to that same chemotherapy protocol did not [90].

Overall, these data sustain the efficacy of PARPi in a neoadjuvant setting, in monotherapy, or in association with chemotherapy. In the BrightTNess trial results were not affirmative, however there is a potential for bias interference, such as population number asymmetry. Several trials in the neoadjuvant setting in different contexts, such as breast, prostate, and ovarian, with variated associations (e.g., hormonotherapy, CDK 4/6 inhibitors, etc.) are ongoing [91,92,93,94,95,96]. Nevertheless, there are no data on the use of PARPi in the neoadjuvant setting in NSCLC.

## 4. Combination of ICI and PARPi

A vital mechanism underlying cancer immune evasion is the expression of inhibitory ligands on the surface of cancerous cells, PD-L1 being the most well-known. Historically, PD-L1 has been identified as the first biomarker for ICIs response, despite that it is an imperfect predictor of ICI response. The serine/threonine protein Glycogen Synthase Kinase 3-beta (GSK3β) is an enzyme that controls glycogen metabolism, interacts with PD-L1, and modulates its expression by inducing proteasome degradation of PD-L1. PARPi has been associated with an increase in PD-L1 expression mainly due to the inactivation of GSK3β, in a dose-dependent manner, suppressing T-cell activation and increasing tumor cell killing [68,97].

Another pathway through which PARPi upregulates PD-L1 is with ATM-ATR-Checkpoint kinase 1 (CHEK1). ATM acts as a kinase sensor for DSBs, after ATM is activated a switch in a signal kinase from ATM to ATR occurs. Finally, the ATM-to-ATR switch activates CHEK1 which further leads to Janus Kinase/Signal Transducer and Activator of Transcription proteins (JAK/STAT) signaling activation and upregulation of PD-L1 expression [68,98].

DNA damage and deficient mechanisms of DNA repair alter the intrinsic immunogenicity, through modulation of surface phenotype and intracellular pathways, but also by modifying the extrinsic immunogenicity of the TME, which plays a major role in response to therapies [68,97]. Tumors with defects in DNA reparation mechanisms can have sustained low-level DNA damage, that promotes inflammation and T_H_1 immune response, leading to extrinsic tumor suppression, due to infiltration of suppressive immune cells, like Myeloid-Derived Suppressor Cells (MDSCs) or Tumor-Associated Macrophages (TAMs), leading to additional DNA damage via free radical release. These mechanisms lead to chronic inflammation, immunosuppression, and cancer progression. PARP regulates T-cell function, maturation, and differentiation, but also DCs recruitment and functioning, macrophage polarization, and increased MDSCs recruitment to TME. PARPis may have the potential to shift from this chronic inflammatory status to a more immune-responsive TME through extensive effects on cells involved in innate and adaptive immune response and soluble factors, resulting in activation of immunosuppressive pathways thus offering targetable immunological vulnerabilities [97].

Another mechanism of interaction between PARPis and the immune system is through the pathway of STimulator of INterferon Genes (STING), a system involved in the production of IFN-γ and pro-inflammatory cytokines. Genomic instability in tumor cells, either induced by PARPi or innate mutations, leads to the accretion of incompletely repaired DNA, producing tumor-derived double-strand DNA (dsDNA) in the cytoplasm. The dsDNA can be detected by cytosolic DNA sensor cyclic GMP-AMP Synthase (cGAS) that leads to activation of the STING signaling pathway. STING stimulates phosphorylation and nuclear translocation of IFN I transcriptional regulatory factors TANK- binding kinase 1 (TBK1) and IFN regulatory factor 3 (IRF3), it also activates NF-κB pathway which acts with IRF3. IFN I upregulation promotes systemic immune response and regulates multiple components in anticancer immunity, especially T cells, NK cells, and DCs. IFN also stimulates the JAK/STAT pathway which leads to the expression of IFN-related genes. STING upregulation also leads to PD-L1 expression [68,97,98,99].

In many solid tumors, a mutation in genes concerning DNA repair, either innate or acquired, enhances TMB and neo-antigens production, and this has been strongly correlated with ICI responses, even though the optimal TMB cut-off remains unclear across tumor types. Furthermore, TMB further increases neo-antigens production and TILs due to a greater expression of genes involved in immune response, such as the TCR signaling, IFN-γ, and TNF receptors. PARPi induces DNA damage and results in DNA fragment accumulation within the cytoplasm. Thus, resulting in a higher expression of neoantigens exposed in the cell surface, increasing immune response activation and TMB improving immunogenicity, and therefore a better potential response for ICI [68,97,99].

In summary, PARPi leads to the amplification of STING signaling; upregulation of interferons and chemoattractants; the repertoire of tumor antigens; TMB; T cell activation and recruitment; and the expression of immune blockade targets, like PD-L1 [38,68,69]. This is expected to prime the TME making it more susceptible to immunotherapy and offering a chance for a more robust and durable response [38,68,69,97,100,101,102]. The mechanism of ICI and PARPi interaction is mechanism shown in Figure 4.

### ICI and PARPi Combination Trials

Thus, a therapeutical combination of PARPi and other agents, to improve its effects and overcome resistances, is being explored in many tumor sets. Initial studies combined PARPi with chemotherapy, radiotherapy, TKIs, or other regimens. However, current approaches are based on PARPi combination with immunotherapy, in different types of tumors, with very promising results and evidence of synergy [38,68,97,98,99,103]. Several trials with a combination of PARPi and ICI are currently ongoing in lung cancer, genitourinary, gastrointestinal, gynecological, and other areas [68]. Table 1 summarizes the most important data from the reviewed trials.

One of the first trials to show the benefit of the association between PARPi and ICI in solid tumors was the **TOPACIO trial**, published in June 2018. This phase II trial combines pembrolizumab and niraparib in recurrent ovarian and advanced TNBC. In the ovarian cancer group the ORR was 18% with a Disease Control Rate (DCR) of 65%, nearly one-third of the patients were platinum-resistant and these presented an ORR of 25%. The TNBC patients had an ORR of 21% and a DCR of 49% [104,105]. 

The phase II basket **MEDIOLA** trial, published in October 2018, included a subgroup with SCLC, the patients were treated with a combination of durvalumab (anti-PD-L1) and Olaparib. From the available data results, patients with platinum-sensitive relapsed ovarian cancer with BRCA mutation showed an ORR of 63%, DCR at 12 weeks of 81%, and median PFS (mPFS) of 11.1 months in the association arm. In patients with ovarian cancer BRCA wild-type, the ORR was 31.3%, DCR 28.1%, and mPFS 5.5 months. The subgroup with relapsed gastric cancer showed an ORR of 10% and a DCR of 26%. In patients with SCLC, the DCR was 29% [106,107]. 

**Table 1 ijms-24-04044-t001:** Trials PARP inhibitors plus ICI, summative information [68,104,105,106,107,108,109].

Trial	NCT02657889 (TOPACIO/ KEYNOTE-162)	NCT02734004 (MEDIOLA)	NCT03308942 (JASPER)	NCT03330405 (JAVELIN PARP Medley)
**Phase**	I-II	II	II	Ib-II
**Status**	Complete	In progress	Complete	Complete
**Drugs**	Niraparib + Pembrolizumab	Olaparib + Durvalumab	Niraparib + Pembrolizumab	Talazoparib + Avelumab
**Tumor types**	TNBC Ovarian cancer	mBC (gBRCAm HER2-) Ovarian cancer SCLC Gastric cancer	**NSCLC** Cohort 1—PD-L1 ≥50% Cohort 2—PD-L1 <50%	**NSCLC** Breast Ovarian Prostate Solid Tumors
**Patients (N)**	55 TNBC 60 Ovarian cancer **Total—115**	34 mBC 32 Ovarian Cancer gBRCAm 32 Ovarian Cancer BRCAwt 39 Gastric cancer 38 SCLC **Total—175**	**16** Cohort 1 **16** Cohort 2 **Total—38**	**42 NSCLC** **5 DDR + NSCLC** 23 HR+, HER2–, DDR+ mBC 22 TNBC 20 Ovarian cancer 11 Ovarian cancer BRCAm **Total—123**
**Outcomes**	**Ovarian:** ORR 18% DCR 65% PFS 3.4m sAE 37.7% irsAE 6% **TNBC** ORR 21% DCR 49% PFS 2.3m sAE 20% irsAE 4%	**mBC:** ORR 56% DCR 47% PFS 6.7m sAE 32% **Ovarian gBRCAm:** ORR 63% DCR 81% PFS 11.1m **Ovarian non-gBRCAm:** ORR 31.3% DCR 28.1% PFS 5.5m **Gastric:** ORR 10% DCR 26% sAE 48% **SCLC:** DCR 29% sAE 34.2%	**Cohort 1** ORR 56,3% DCR 87% PFS 8.4m OS NR sAE 88.2% **Cohort 2** ORR 20% DCR 70% PFS 4.2m OS 7.7m sAE 85.7%	**NSCLC** ORR 16.7% **DDR + NSCLC** ORR 20% **TNBC** ORR 18.2% DoR 11.1m **DDR+ TNBC** ORR 34.8% DoR 15.7m **Ovarian** ORR 20% **BRCAm Ovarian** ORR 63.6% DoR NR **All tumors** sAE—33.6%

BRCAwt—BRCA wild type, DCR—Disease Control Rate, DDR—DNA damage response positive, DoR -Duration of response, gBRCAm—germinative BRCA mutation, HER-2—Human Epidermal growth factor Receptor 2, irsAE—immune-related serious Adverse Events, m—months, mBC—metastatic Breast Cancer, NR—Non- Reached, NSCLC—Non-Small Cell Lung Cancer, ORR—Overall Response Rate, PFS—Progression Free Survival, sAE—serious Adverse Events, SCLC—Small Cell Lung Cancer, TNBC—Triple Negative Breast Cancer.

In the **JASPER** phase II trial, published in January 2022, the association of pembrolizumab and niraparib was studied in patients with advanced-stage NSCLC. The patients were stratified according to the expression of PD-L1 Tumor Proportion Score (TPS) ≥50% (cohort 1) or 1–49% (cohort 2). In cohort 1, ORR was 56.3%, whereas in cohort 2, ORR was 20.0%. In cohorts 1 and 2, the median Duration of Response (mDoR) was 19.7 months and 9.4 months, mPFS 8.4 months and 4.2 months, and the mOS was NR (non-reached) and 7.7 months, respectively. Grade ≥3 AEs occurred in 88.2% and 85.7% of patients in cohorts 1 and 2, respectively. Safety was consistent with the known profiles of these agents in monotherapy [108].

In the most recent study, published in November 2022, a phase Ib/II basket nonrandomized controlled trial—**JAVELIN PARP Medley**—patients with advanced solid tumors, including NSCLC, metastatic Breast Cancer (mBC), ovarian, were treated with an association of talazoparib plus Avelumab. In NSCLC the ORR was 16.7%, mDoR of 17.5 months, in the subgroup with DNA damage response (DDR)-positive ORR was 20% and mDoR NR. This association was generally well tolerated and no new safety concerns were identified [109].

Several trials are testing the efficacy of ICI and PARPi combination in neoadjuvant setting in different tumor locations, mainly in breast cancer with BRCA mutations. However, so far, no results from these trials are available and also there are no trials including NSCLC ongoing [110,111,112,113,114,115].

## 5. Conclusions

In the past, strategies to improve immunotherapy responses, and subsequently oncologic outcomes, were based on biomarker selection, like PD-L1 expression. However, this approach only benefits a specific set of patients. Therefore, in other to improve immunotherapy efficacy, overcome resistances and extend this potential benefit to other populations, studies have been focusing on strategies that combine different treatment modalities, such as immunotherapy with chemotherapy, radiotherapy, TKI, or other therapy combinations.

At present, one of the main foci of investigation has been the association of ICIs with DNA-damaging agents, such as PARPi. The rationale for the additive activity of both pharmacologic classes, described in this paper, is based on mechanisms of action already recognized and supported by clinical evidence. Hence, considering that immunotherapy only provides help for some patients, and that early-stage NSCLC still has a high risk of relapse and progression despite the current standard of care treatment (surgery ± chemotherapy ± radiotherapy). Development of clinical trials to access the efficacy and safety of neoadjuvant combined therapy with ICI and PARPi in patients with resectable early-stage NSCLC is a priority. These data will allow us to study newer, and possibly more effective, treatment strategies that might improve the oncologic outcomes of these patients.

## Figures and Tables

**Figure 1 ijms-24-04044-f001:**
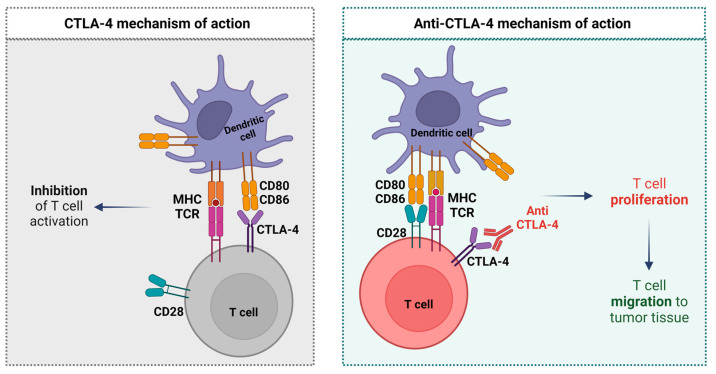
Anti-CTLA-4 mechanism of action (image adapted from [37,38]).

**Figure 2 ijms-24-04044-f002:**
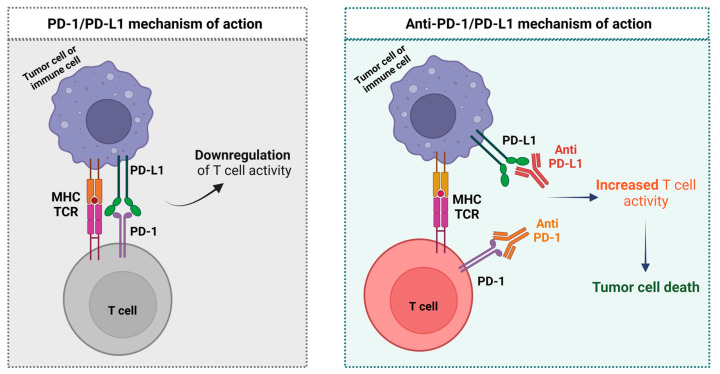
Anti-PD-1/PD-L1 mechanism of action (image adapted from [37]).

**Figure 3 ijms-24-04044-f003:**
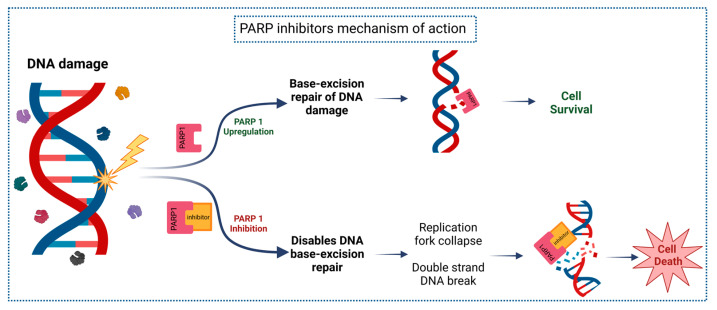
PARP inhibitors mechanism of action (image adapted from [75]).

**Figure 4 ijms-24-04044-f004:**
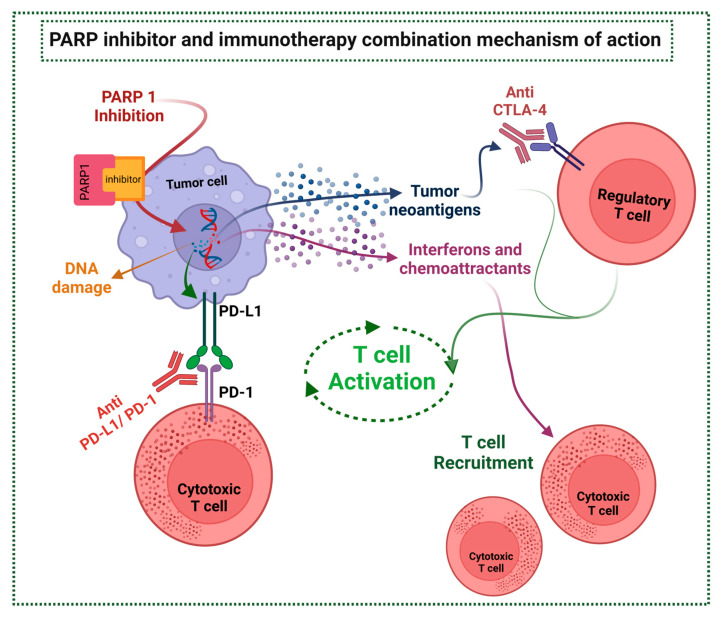
PARP inhibitor and immunotherapy combination mechanism of action (image adapted from [37,38,75]).

## Data Availability

All data included in this study was provided from Clinical trials database and is available.

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
