# Peer review of "New Approaches in Early-Stage NSCL Management: Potential Use of PARP Inhibitors and Immunotherapy Combination"

_ijms, 2023, doi:10.3390/ijms24044044_

Round 1

Reviewer 1 Report

The topic of the review is extremely interesting, but unfortunately there are no data regarding neaadjuvant therapy in NSCLC with PARP inhibitors and immunotherapy combination, therefore it is not possible to make a review on this topic.

The article is well written and precisely analyzes the mechanisms of action of the drugs. Basically, the paper has a logic and clear structure, and has a clear message. The quality of English scientific is good. The abstract is concise and summarizes the essential information of the paper. The bibliography is up to date. Unfortunately, the title is not appropriate for the content of the article.

In order not to lose the work done, I advise the authors to turn the article into a review on the PARP inhibitors and immunotherapy combination in advanced NSCLC. It would be very interesting and with more solid data to report.

Reviewer 2 Report

Thank you for the opportunity to review your paper.

The author describes the future prospects for the combination of PARP inhibitors and immunotherapy in the perioperative treatment of lung cancer.

I think that the section 1. is redundant. The section on 1. and 2. may be integrated. Moreover, the section 2. should focus on the current status of perioperative treatment using ICI regimens.

Combination therapy with PARP inhibitors and ICI is being tried as an expansion of treatment for advanced stage lung cancer, but is there a basic background for this in patients with the perioperative lung cancer? Moreover, in cancer types where PARP inhibitors have already been approved, what is the status of perioperative development of therapies that include PARP inhibitors?

Reviewer 3 Report

1. The immunotherapy response and PARPi treatment have been found to be correlated with tumor mutation burden and the production of neo-antigens. References for this information should be included in Figure 4.

 2. The association between PARPi response and mutation hotspots in early-stage Non-Small Cell Lung Cancer was not investigated by the authors.

 3. Some relevant references on cancer therapy are suggested,

e.g.

Cancers (Basel). 2022 Oct 28;14(21):5305. doi: 10.3390/cancers14215305. PMID: 36358724; PMCID: PMC9654807.

 Cancer Med. 2019 May;8(5):2179-2187. doi: 10.1002/cam4.2120. Epub 2019 Apr 2. PMID: 30941903; PMCID: PMC6536970.

 Front Oncol. 2022 Jul 22;12:819555. doi: 10.3389/fonc.2022.819555. PMID: 35936696; PMCID: PMC9354680.

Round 2

Reviewer 2 Report

I think all issues have been addressed.